# MULTIMODAL DATASETS WITH CONTROLLABLE MUTUAL INFORMATION

## ABSTRACT

We introduce a framework for generating highly multimodal datasets with explicitly calculable mutual information between modalities. This enables the construction of benchmark datasets that provide a novel testbed for systematic studies of mutual information estimators and multimodal self-supervised learning techniques. Our framework constructs realistic datasets with known mutual information using a flow-based generative model and a structured causal framework for generating correlated latent variables.

## 1 INTRODUCTION

Self-supervised learning (SSL) has become a core component of many state-of-the-art large-scale machine learning models (Balestriero et al., 2023). Such models are also increasingly *multimodal*, i.e. designed to learn from varied input sources such as text, images, and audio (Radford et al., 2021; Zong et al., 2024). A prevailing intuition is that multimodal SSL is effective because different modalities provide complementary "views" of the same underlying concepts, enabling the learning process to exploit their shared information. The precise relationship between the mutual information (MI) between modalities and SSL performance, however, is not fully understood.

Contrastive SSL methods built using the InfoNCE loss function (Oord et al., 2018; Chen et al., 2020; He et al., 2020; Caron et al., 2020) have a clear information-theoretic interpretation: for example, the learned similarity scores estimate the pointwise mutual information (PMI) between paired samples. By contrast, no analogous theoretical connection has been established for either highly multimodal settings (i.e. $N > 2$ modalities) or for non-contrastive SSL methods such as multimodal masked modeling (Mizrahi et al., 2023; He et al., 2022; Wang et al., 2023; Huang et al., 2025), despite the fact that both of these directions are quickly gaining prominence in the field (Li et al., 2024; Hondru et al., 2025). A theoretically-grounded understanding of the fundamental relationship between inter-modality mutual information and SSL representations (and their corresponding performance on downstream tasks) will be increasingly critical as models continue to scale to larger numbers of input modalities. In particular, principled frameworks will be needed to evaluate how the distribution of shared information across modalities influences the quality of the learned embeddings.

Complicating matters further, MI is notoriously difficult to estimate from samples, particularly in high-dimensional, real-world datasets (McAllester & Stratos, 2020; Czyż et al., 2023a). A wide range of MI estimators have been proposed using techniques such as kernel estimation, $k$-nearest neighbor, and neural estimators (Pizer et al., 1987; Kozachenko & Leonenko, 1987; Moon et al., 1995; Kraskov et al., 2004; Belghazi et al., 2018; Song & Ermon, 2020b; Butakov et al., 2024; Belghazi et al., 2021). However, these estimators are typically only validated on synthetic datasets of simple distributions for which the MI is analytically tractable (Darbellay & Vajda, 1999; Suzuki, 2016; Czyż et al., 2023a;b; Butakov et al., 2023).

Datasets with controllable MI that emulate the challenges of real-world data are needed to better understand the advantages, disadvantages, and tradeoffs of different SSL learning objectives. Such datasets can enable systematic, reproducible studies of how multimodal SSL representations depend on information overlap and shared features, offering both theoretical insights and practical guidance for model design. Finally, they provide a reliable testbed for evaluating the performance of various mutual information estimation strategies designed for use on real-world datasets.

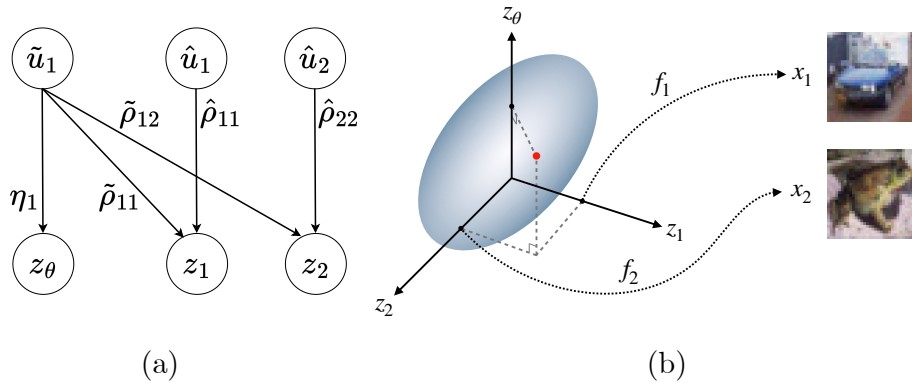

(a)            (b)

Figure 1: **Schematic of our dataset generation framework.** a) An example DAG showing the linear mixing of proto-latents $\mathbf{u}$ via coefficients $\eta, \rho$ into interpretable correlated Gaussian latent variables $\mathbf{z}$. b) Overview of sampling from a multidimensional Gaussian to draw latent inputs $z_1$, $z_2$, and $z_\theta$ that are fed into invertible maps $f_1$ and $f_2$ to a realistic feature space.

In this work, we introduce a framework to generate realistic multimodal data with controllable mutual information. Figure 1 shows an overview of our data generation framework:

(a) First, we use directed acyclic graphs (DAGs) to generate easily interpretable correlated Gaussian latent variables $\mathbf{z}$ with known mutual information.

(b) We then feed the outputs $\mathbf{z}$ of these DAGs into invertible bijective transformations to construct multimodal datasets where the amount and distribution of shared information can be explicitly controlled across multiple modalities.

## 2 BACKGROUND

**Mutual Information (MI).** Mutual information $I(X; Y)$ is a fundamental quantity from information theory that measures the statistical dependence between two random variables $X$ and $Y$. It is formally defined as the Kullback-Leibler (KL) divergence of the joint distribution $p(X, Y)$ and the product of the marginal distributions $p(X)p(Y)$:

$$I(X; Y) \equiv D_{KL}\left(p(X, Y) \,||\, p(X)p(Y)\right)$$

Alternatively, it can also be expressed in terms of the Pointwise Mutual Information (PMI):

$$I(X; Y) \equiv \mathbb{E}_{x,y \sim p(X,Y)}\left[\text{PMI}(x; y)\right]$$

The MI quantifies the extent to which one variable reduces uncertainty in the other. For instance, when $X$ and $Y$ are fully independent, their joint distribution $p(X, Y)$ reduces to the product of the marginal distributions $p(X)p(Y)$, therefore the MI is zero.

**Pointwise Mutual Information (PMI).** When evaluated on specific values $x \sim p(X)$ and $y \sim p(Y)$, the pointwise MI (PMI) captures the probability of these two values occurring together compared with that same probability if they were fully independent. The PMI is formally expressed as:

$$\text{PMI}(x; y) \equiv \log\left(\frac{p(x, y)}{p(x)p(y)}\right).$$

**Multimodal Self-Supervised Learning (SSL).** Often compared to the primary human senses such as sight, hearing, or touch, modalities in machine learning refer to distinct forms of sensing the world and the corresponding representations of the observed data. Modalities can have radically different formats (e.g. RGB images and timeseries data), or they can exhibit similar formats but describe distinct information sources (e.g. RGB images and segmentation maps). In this paper, our operational definition for a modality is a random variable $X_m$ and a corresponding sample space

$\mathcal{X}_m$. In particular, if $X_1$ and $X_2$ correspond to two different distributions, then we consider them to be two separate modalities regardless of their data format. Multimodal SSL often involves learning a joint representation of many data modalities, which we view as distinct from multi-*view* SSL, which generally consists of learning a joint representation of multiple views derived from the same data modality, e.g. different crops of a single image. Multimodal SSL uses the relationships between modalities to learn joint representations without explicit labels.

**Flow-based generative modeling.** Flow-based generative models are designed to facilitate the direct transformation between probability distributions using invertible mappings applied to a simple base distribution such as a Gaussian. Each transformation is designed to be bijective with a tractable Jacobian determinant, enabling exact computation of both likelihoods and samples. Because they provide both efficient sampling and exact density evaluation, flow-based models are increasingly used not only in generative modeling but also in scientific applications requiring tractable likelihoods and explicit control over distributions. *Flow-matching* (Albergo & Vanden-Eijnden, 2022; Lipman et al., 2024) is a recent approach within the family of flow-based generative models that enables efficient training of Continuous Normalizing Flows (CNFs) (Chen et al., 2018a) by directly regressing the velocity field that transports a base distribution to the data distribution instead of optimizing the exact maximum-likelihood objective.

## 3 CREATING DATASETS WITH CONTROLLED MUTUAL INFORMATION

Our goal is to enable rigorous, scalable experiments using multimodal datasets where the MI between modalities is precisely specified and easy to interpret. To accomplish this, we design an expressive three-step framework $\mathbf{u} \rightarrow \mathbf{z} \rightarrow \mathbf{x}$. This begins with uncorrelated, normally distributed 'proto-latent' variables $\mathbf{u}$, which are related by linear structural equations to form an easy-to-interpret causal model for latent variables $\mathbf{z}$, for which mutual information is easy to compute. Finally, we use blocks of components of $\mathbf{z}$ as the input to a set of invertible transformations $\{f_i\}_{i=1}^n$ (one for each of $n$ modalities) to produce synthetic observations $\mathbf{x}_i = f_i(\mathbf{z}_i)$ that preserve the mutual information between the corresponding latent variables. In this work, we implement $f_i$ as flow-matching models that have been pretrained to produce realistic images.

In addition to the $\mathbf{x}_i$ for each modality, we also generate a (scalar) target variable $\theta$ computed from the latent variable $z_\theta$. We partition the vector of proto-latents into sets of components $\mathbf{u} = (\tilde{\mathbf{u}}, \hat{\mathbf{u}})^T$. The goal here is to isolate a source of randomness $\tilde{\mathbf{u}}$ that can be interpreted as a common cause that induces correlation between the observed $\mathbf{x}_i$ and some target quantity of interest $\theta$ that one may wish to estimate from the $\mathbf{x}_i$. For simplicity, we take $\theta$ to be a scalar and let $\theta = z_\theta$ since complicated non-linear relationships between $\mathbf{x}_i$ and $\theta$ are already captured by the flows $f_i$.

### 3.1 GENERALIZED LINEAR CAUSAL CONSTRUCTION: PROTO-LATENT TO LATENT CONNECTIONS

We wish to create a large latent variable vector $\mathbf{z} = (z_\theta, \mathbf{z}_1, \dots, \mathbf{z}_{N_z})^T$ that is distributed according to a multivariate Gaussian with known covariance for which the mutual information is easy to compute. We achieve this by forming linear combinations of i.i.d. normally distributed proto-latents $\mathbf{u} \sim \mathcal{N}(\mathbf{0}, \mathbf{I})$:

$$\mathbf{z} = \begin{pmatrix} z_\theta \\ \mathbf{z}_1 \\ \vdots \\ \mathbf{z}_{N_z} \end{pmatrix} = \mathbf{A} \begin{pmatrix} \tilde{\mathbf{u}} \\ \hat{\mathbf{u}}_1 \\ \vdots \\ \hat{\mathbf{u}}_{N_u} \end{pmatrix} \tag{1}$$

where:

- $\tilde{\mathbf{u}} \in \mathbb{R}^{N_\theta}$: proto-latents serving as a common cause inducing correlation between $\theta$ and $\mathbf{x}_i$,

- $\hat{\mathbf{u}}_j \in \mathbb{R}^d$: proto-latents for the observed modalities, $j = 1, \dots, N_u$,

- $z_\theta \in \mathbb{R}$: scalar target quantity of interest (e.g., a physical quantity to be estimated from $\mathbf{x}_i$),

- $\mathbf{z}_i \in \mathbb{R}^d$: latent variables associated to each observed modality, $i = 1, \ldots, N_z$,
- $\mathbf{A}$: a user-defined matrix specifying the structural equations in the causal model relating $\mathbf{u}$ and $\mathbf{z}$.

The matrix $\mathbf{A}$ encodes all structured dependencies between latent variables and outputs. One could use an arbitrary matrix $\mathbf{A}$, but that would lack interpretability. Instead, we structure $\mathbf{A}$ to follow from an expressive, easy-to-interpret causal story.

For example, the causal model shown in Fig.1(a) corresponds to the structural equations

$$
\begin{aligned}
z_\theta &= \eta \tilde{u}_1 \\
\mathbf{z}_1 &= \tilde{\rho}_{11} \tilde{u}_1 \, \mathbf{1}_d + \hat{\rho}_{11} \hat{\mathbf{u}}_1 \\
\mathbf{z}_2 &= \tilde{\rho}_{12} \tilde{u}_1 \, \mathbf{1}_d + \hat{\rho}_{22} \hat{\mathbf{u}}_2 \ ,
\end{aligned}
\tag{2}
$$

where $\mathbf{1}_d$ is a $d$-dimensional vector of ones. The hyperparameters of this model are $\eta, \tilde{\rho}_{ki}, \hat{\rho}_{ji} \in \mathbb{R}$. Note we treat $\tilde{\mathbf{u}}$ and $\hat{\mathbf{u}}$ asymmetrically: $\tilde{\mathbf{u}}$ is a common cause that feeds into both $z_\theta$ and the latents associated to the individual modalities, while $\hat{\mathbf{u}}$ does not feed into $z_\theta$. In this simple example, $\hat{\mathbf{u}}$ is also the only source of correlation between $z_1$ and $z_2$ – and the mutual information between $z_1$ and $z_2$ is perfectly predictive of $z_\theta$.

In Section 4 we will consider other causal stories, their corresponding linear structural equations, and the consequences of these relationships on the induced mutual information between $\theta$ and $\mathbf{x}_i$.

## 3.2 Analytic Mutual Information of the Latent Variables

We provide a derivation of the mutual information calculation between latent variables $\mathbf{z}$ constructed as described in Section 3.1. The covariance matrix of the latents is simply given by:

$$
\Sigma = \mathrm{Cov}(\mathbf{Z}, \mathbf{Z}) = \mathbf{A}\mathbf{A}^\top
\tag{3}
$$

We can represent the covariance matrix in block form corrresponding to $z_\theta$, $\mathbf{z}_1$, and $\mathbf{z}_2$ as

$$
\Sigma = \begin{bmatrix} \Sigma_{\theta\theta} & \Sigma_{\theta 1} & \Sigma_{\theta 2} \\ \Sigma_{1\theta} & \Sigma_{11} & \Sigma_{12} \\ \Sigma_{2\theta} & \Sigma_{21} & \Sigma_{22} \end{bmatrix}
\tag{4}
$$

For any two blocks in $\Sigma$, we define the reduced block matrix:

$$
\Gamma_{ij} = \begin{bmatrix} \Sigma_{ii} & \Sigma_{ij} \\ \Sigma_{ji} & \Sigma_{jj} \end{bmatrix}
\tag{5}
$$

For multivariate Gaussian distributions, the mutual information is a simple function of the determinants of these block covariance matrices. For example,

$$
I(\theta; Z_1) = \frac{1}{2} \ln \left( \frac{|\Sigma_{\theta\theta}||\Sigma_{11}|}{|\Gamma_{\theta 1}|} \right)
\tag{6}
$$

$$
I(Z_1; Z_2) = \frac{1}{2} \ln \left( \frac{|\Sigma_{11}||\Sigma_{22}|}{|\Gamma_{12}|} \right) \ ,
\tag{7}
$$

where $| \cdot |$ denotes the determinant of the corresponding block covariance matrix.

While the covariance matrix and mutual information quantities in the preceding equations can be calculated numerically, we are also able to derive closed-form, analytical equations for various mutual information quantities (see Appendix B). One benefit of the closed form solutions is they reveal scaling in terms of the hyperparameters of the structural equations, the number of modalities, and the dimensonality of each modality. In the case of the causal model considered in Fig. 1(a) and Eq. 2, we find

$$I(\theta; Z_1) = -\frac{1}{2} \log \left( 1 - \frac{d\, \tilde{\rho}_{11}^2}{\hat{\rho}_{11}^2 + d\, \tilde{\rho}_{11}^2} \right) \tag{8}$$

$$I(Z_1; Z_2) = -\frac{1}{2} \log \left( 1 - \frac{d^2\, \tilde{\rho}_{11}^2 \tilde{\rho}_{12}^2}{[\hat{\rho}_{11}^2 + d\, \tilde{\rho}_{11}^2][\hat{\rho}_{22}^2 + d\, \tilde{\rho}_{12}^2]} \right) . \tag{9}$$

These equations that we have derived have been verified against the numerical calculations.

### 3.3 Flow-based generative modeling preserves mutual information

The final step of our three-step process is to create realistic synthetic data in multiple modalities. Recall that the latent vector is organized by blocks of components as $\mathbf{z} = (z_\theta, \mathbf{z}_1, \ldots, \mathbf{z}_{N_l})^T$. We transform the individual blocks of latent variables independently, yielding $\mathbf{x}_i = f_i(\mathbf{z}_i)$, where the $f_i(\cdot)$ are generative models pretrained on real-world datasets.

We leverage a key result that states that if the $f_i$ are continuous bijective maps, then the mutual information is preserved:

$$I(X_i; X_j) = I(Z_i; Z_j) . \tag{10}$$

This result can be seen as following from the data-processing inequality and is also the result of a direct computation of the mutual information after a change of variables, where the Jacobian factors that arise cancel exactly (see e.g., Cover & Thomas, 2006; Czyż et al., 2023a;b). While many generative models satisfy this condition, e.g., discrete-time normalizing flows (Rezende & Mohamed, 2015; Papamakarios et al., 2021), we use continuous-time normalizing flows based on flow matching (Lipman et al., 2024; Albergo & Vanden-Eijnden, 2022; Liu et al., 2022) in this work. We pretrain on CIFAR-10 (Krizhevsky, 2009) using image class as a proxy for modality (i.e., $f_0$ is trained on images of cars, $f_1$ is trained on images of frogs), but we emphasize that our framework is agnostic to $f_i$ parameterization and modality definition.

### 3.4 Templates

The mutual information $I(X_i, X_j)$ does not specify how this information is distributed across the components of $X_i$ and $X_j$. Similarly, the pointwise mutual information in two images does not uniquely determine the spatial location of their correlated pixels. Nevertheless, the way the information is distributed matters in practice because architectural choices are sensitive to those details. The impact of these architectural choices on the performance of competing approaches to SSL or mutual information estimation then become conflated other algorithmic choices (e.g. data augmentation and training objectives) that are more clearly tied to (pointwise) mutual information.

Ideally, we would like to perform ablation studies designed to disentangle these effects. This requires being able to independently vary the mutual information and the way that information is distributed across the components of the random variables. In order to achieve this, we introduce the notion of *templates* into our $\mathbf{u} \to \mathbf{z}$ mapping.

We define a template $\mathbf{T}_{ik} \in \mathbb{R}^d$ as a linear map relating the common cause $\tilde{u}_k$ and the a latent $\mathbf{z}_i$:

$$\mathbf{z}_i = \sum_{k=1}^{N_\theta} \tilde{u}_k \mathbf{T}_{ik} + \sum_{j=1}^{N_u} \hat{\mathbf{u}}_j \tag{11}$$

For example, $\mathbf{T}_{ik} = \frac{1}{d}\mathbf{1}_d$ implements a homogeneous distribution of information about $\tilde{u}_k$ across the latent $\mathbf{z}_i$, while $\mathbf{T}_{ik} = (0, \ldots, 0, 1, 0, \ldots, 0)^T$ implements a scenario where all the information about $\tilde{u}_k$ is concentrated in a single component of $\mathbf{z}_i$.

This design is motivated by real-world scenarios in which the information about multiple common causes is distributed nonuniformly across several modalities (e.g., multiple supernovae being imaged by multiple types of telescopes). With templates, future studies can better understand the impact of architectural design choices based on the information distribution in various modalities.

## 4 EXAMPLES OF DATASETS RESULTING FROM OUR MODEL

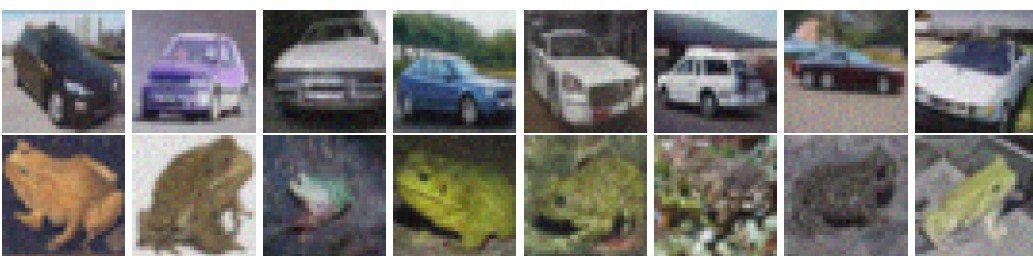

Figure 2: Eight examples of correlated pairs of images $(\mathbf{x}_1, \mathbf{x}_2)$ generated from our procedure representing two realistic modalities. In this case, both modalities are images, but corresponding to flows conditioned on different class labels ("automobile", "frog") from CIFAR-10 (Krizhevsky, 2009). The dimensionality of the data in both cases is $d = 32 \times 32 \times 3 = 3072$.

As mentioned in Sec. 3.1, the matrix $\mathbf{A}$ allows for an arbitrary linear structural equation between the proto-latents $\mathbf{u}$ and the latent variables $\mathbf{z}$. While this flexibility may come at the cost of interpretability, we find that in fact many realistic causal stories are well captured by structural equations with only a few hyperparameters.

We show two specific examples in this section. All examples are implemented using a (conditional) flow matching model pretrained on CIFAR-10 data, where image class label is used as a proxy for different modalities. Figure 2 shows eight examples of correlated pairs $(\mathbf{x}_1, \mathbf{x}_2)$ generated from our procedure. **While there is no clear visual connection between these pairs of images, our framework allows us to state unequivocally that these high-dimensional, complex image pairs have a specific quantity of mutual information – a feat that was previously unattainable.**

### 4.1 EXAMPLE 1: EMPIRICALLY DEMONSTRATING EXAMPLE USE CASES

**Benchmarking mutual information estimators.** We demonstrate the use of our framework to generate a benchmark dataset to explore the performance of a number of popular mutual information estimators. We use the causal model shown in Figure 1 to generate a set of ten datasets with mutual information $I(X_1; X_2)$ ranging from 0.0284 to 1.39, with each dataset consisting of 10,000 paired CIFAR-like images. We use an existing benchmark suite (Lee & Rhee, 2024) to estimate the empirical mutual information and compare it to the ground-truth mutual information in Figure 3. We report the correlation and RMSE for each estimator in Table 1. We observe that the regressed mutual information from all the estimators follows the linearly increasing ground truth mutual information.

| MI Estimator | Correlation | RMSE |
|---|---|---|
| DV (Donsker & Varadhan, 1983) | 0.995 | 0.1094 |
| JS (Nowozin et al., 2016) | 0.993 | 0.3049 |
| InfoNCE (Oord et al., 2018) | 0.991 | 0.1981 |
| MINE (Belghazi et al., 2018) | 0.993 | 0.0983 |
| NWJ (Nguyen et al., 2010) | 0.996 | 0.0851 |
| SMILE-1 (Song & Ermon, 2020a) | 0.998 | 0.0935 |
| SMILE-5 (Song & Ermon, 2020a) | 0.993 | 0.0974 |
| SMILE-inf (Song & Ermon, 2020a) | 0.993 | 0.0969 |

Table 1: Correlation and RMSE from linear regression of estimated vs. ground-truth mutual information, computed for a number of mutual information estimators.

**Regressing the target variable $\theta$ from $X$.** We evaluate how well a model is able to regress the target variable $\theta$ from data $X_1$ as a function of the mutual information $I(\theta, X_1)$. Intuitively, models with fixed capacity should predict $\theta$ more accurately from data that contain more information about $\theta$, i.e. larger $I(\theta; X_1)$. We re-use the same ten datasets, for which $I(\theta; X_1)$ ranges from 0.134 to

1.73. For each dataset, we regress $\theta$ from the images using a shallow convolutional network, choosing the best model after training for $500$ epochs. We show that the best achievable RMSE decreases with increasing mutual information between $X_1$ and $\theta$ (Figure 4) and show example distributions of prediction error $\theta - \hat{\theta}$ for two representative MI values in Appendix A.

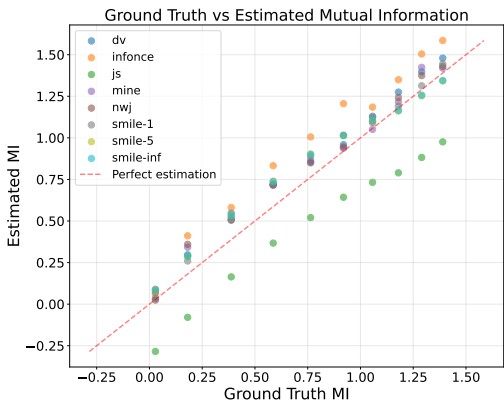
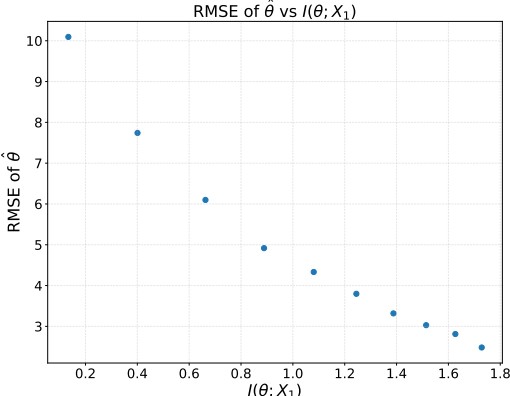

Figure 3: MI estimators reliably recover the ground-truth MI from our datasets across a range of MI values.

Figure 4: Models with fixed size and compute budget trained to regress a target scalar $\theta$ monotonically improve with increasing ground-truth MI between the data X1 and $\theta$.

## 4.2 EXAMPLE 2: ESTIMATING A BLACK HOLE'S MASS FROM TWO TELESCOPES

Consider the hypothetical scenario where one wishes to estimate the mass of Sagittarius A*, the supermassive black hole in the center of our Milky Way Galaxy. To do this we might employ two instruments producing two data modalities. Let $\mathbf{x}_1$ represent data from the Event Horizon Telescope, a ground-based array consisting of a global network of radio telescopes. Let $\mathbf{x}_2$ represent data from the Hubble space telescope in orbit around the Earth. Let $\tilde{u}_1$ represent the unknown mass of the black hole and let $\tilde{u}_2$ represent some atmospheric variability that impacts how radio waves propagate in the atmosphere.

The mass of the black hole $\tilde{u}_1$ influences the data from both telescopes; however, the atmospheric effects $\tilde{u}_2$ only impact the data from the Event Horizon Telescope. This narrative is captured by the causal model illustrated in Fig. 5(a). This causal model corresponds to the structural equations

$$
\begin{aligned}
z_\theta &= \eta_1 \tilde{u}_1 + \eta_2 \tilde{u}_2 \\
\mathbf{z}_1 &= \tilde{\rho}_{11} \tilde{u}_1 \mathbf{T}_{11} + \tilde{\rho}_{12} \tilde{u}_2 \mathbf{T}_{12} + \hat{\rho}_{11} \hat{\mathbf{u}}_1 \\
\mathbf{z}_2 &= \tilde{\rho}_{21} \tilde{u}_1 \mathbf{T}_{21} + \hat{\rho}_{22} \hat{\mathbf{u}}_2 \ .
\end{aligned}
\tag{12}
$$

The closed-form, analytical equations for various mutual information quantities corresponding to a similar causal model can be found in Appendix B. The $\mathbf{T}_{ki}$ are *templates* that have the same shape

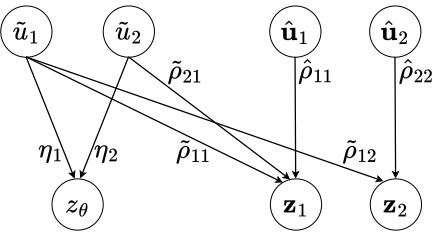
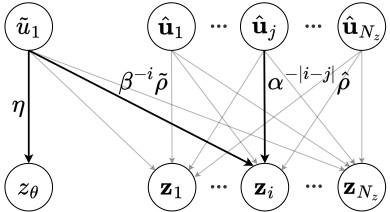

(a) Causal structure for black hole example.

(b) Causal structure for multimodal example.

Figure 5: Examples of causal structures with corresponding linear structural equations that induce specific mutual information.

as the $\mathbf{z}_i$ and can encode some type of inhomogeneous (spatial) structure in the latents. For example, the templates $\mathbf{T}_{11}$ and $\mathbf{T}_{12}$ associated with $\tilde{u}_1$ (black hole mass) are designed to concentrate at the center of the galaxy and dissipates away from the center. Similarly, the template $\mathbf{T}_{21}$ associated with $\tilde{u}_2$ (atmospheric effect) are designed to be diffuse across the whole example. Not shown explicitly in the figure are the functions that generate the observed data from the latents: $\theta = z_\theta$, $\mathbf{x_1} = f_1(\mathbf{z}_1)$, and $\mathbf{x_2} = f_2(\mathbf{z}_2)$.

Fig. 5(a) also has paths from $\tilde{u}_1$ and $\tilde{u}_2$ to $z_\theta$ scaled by the hyperparameters $\eta_1$ and $\eta_2$. This flexibility allows us to capture two different narratives in the same model by changing the values of $\eta_i$. In one narrative, $\theta$ represents the mass of the black hole and corresponds to $\eta_1 = 1, \eta_2 = 0$. In the second narrative, $\theta$ represents the atmospheric effect and corresponds to $\eta_1 = 0, \eta_2 = 1$.

Table 2 shows the result of the mutual information when all of the $\rho$ variables are set to 1 and the dimensionality of the data in each modality is $d = 3072$. Note that in the scenario where the quantity of interest $\theta$ corresponds to the atmospheric effect, that there is no mutual information between the data from the Event Horizon Telescope and the quantity of interest.

Table 2: Mutual information for two scenarios corresponding to the causal structure in Fig. 5(a).

| $\theta$ Represents | $\eta_1$ (Black hole) | $\eta_2$ (Atmosphere) | $I(\theta; X_1)$ | $I(\theta; X_2)$ | $I(X_1; X_2)$ |
|---|---|---|---|---|---|
| Black Hole Mass | 1 | 0 | 2.77 | 0 | 2.63 |
| Atmospheric Effect | 0 | 1 | 2.77 | 3.33 | 2.63 |

### 4.3 EXAMPLE 3: A SCALABLE MODEL FOR MASSIVELY MULTIMODAL DATA

In this example, we shift our emphasis to the number of modalities. The ability to generate correlated tuples of synthetic data $(\mathbf{x}_1, \ldots, \mathbf{x}_{N_z})$ with known mutual information will be extremely valuable for studying the tradeoff among various competing approaches to multimodal SSL. We would like a flexible template that allows us to generate a large number of modalities while keeping a small, fixed number of hyperparameters to reason about. At the same time, we would like the model to be expressive enough to capture some interesting patterns.

We consider the causal model illustrated in Fig. 5(b). This causal model corresponds to the structural equations

$$z_\theta = \eta \tilde{u}_1 \qquad \mathbf{z}_i = \beta^{-i} \tilde{\rho}\, \tilde{u}_1 \mathbf{1} + \sum_{j=1}^{N_u} \alpha^{-|i-j|} \hat{\rho}\, \hat{\mathbf{u}}_j \ . \tag{13}$$

Each set of proto-latents $\hat{\mathbf{u}}_i$ has a corresponding set of latents $\mathbf{z}_i$, which they feed into with a single coefficient $\hat{\rho}$. In addition, the $j^{\text{th}}$ proto-latents also contribute to the $i^{\text{th}}$ latents with some decay constant $\alpha^{-|i-j|}$, with $\alpha \geq 1$. As the hyper-parameter $\alpha$ grows, the correlation between the modalities decays quickly (as a function of $|i-j|$). As $\alpha \to 1$, the modalities become uniformly correlated.

Here we maintain a target quantity of interest for some downstream task (e.g. regression), but only include a single common cause $\tilde{u}_1$. This common cause also induces a correlation among the $\mathbf{z}_i$, but we break the permutation invariance by including a scaling factor $\beta^{-i}$. When $\beta$ is large, only the first few modalities have significant mutual information with $\theta$; however, when $\beta \to 1$, that mutual information with $\theta$ is uniform.

This simple model does not reflect a specific physical scenario, but it does allow for interesting benchmarks and experiments for multimodal SSL. We show in Figure 6 results from training a flow-matching model on 10 CIFAR class labels, allowing us to create these correlated tuples of high-dimensional, realistic images for up to $N_z = 10$. Specifically, we show the mutual information when all of the $\rho$ variables are set to 1, the dimensionality of the data in each modality is $d = 3072$, and various $\alpha$ and $\beta$ are selected. We note that as $\alpha$ increases, $I(X_1; X_i)$ decays at a faster rate and as $\beta$ increases, $I(\theta; X_i)$ decays at a faster rate, as expected. Extending beyond 10 modalities is a straightforward exercise.

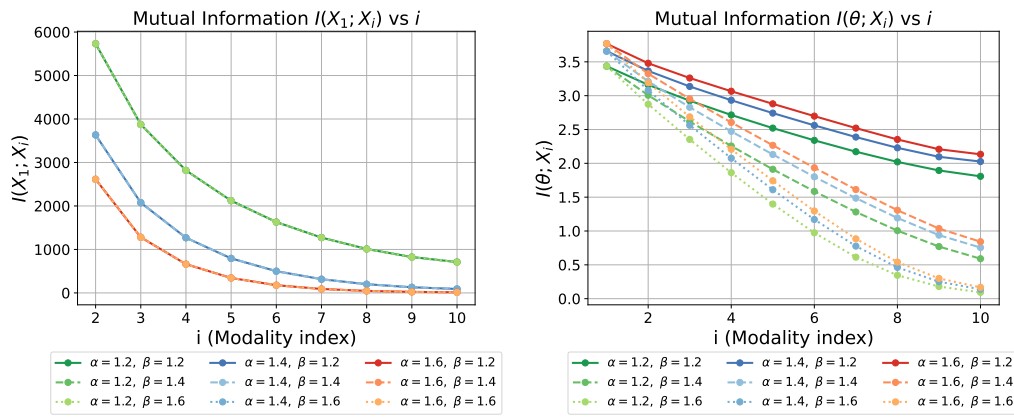

(a) Mutual information between image modalities $X_1$ and $X_i$, for the $i$-th modality.

(b) Mutual information between $\theta$ and image modality $X_i$, for the $i$-th modality.

Figure 6: Information between image modalities $X_1$ and $X_i$ decreases as the distance between $X_1$ and $X_i$ increases. This, as well as the information between image modality $X_i$ and the parameter $\theta$, decreases as the total number of modalities increases.

## 4.4 Example 4: A model for ablation studies for multimodal SSL

While the example in Sec. 4.3 allows one to study the performance of multi-modal SSL methods as a function of the mutual information between the modalities (and the pointwise mutual information between individual samples from those modalities), it does not provide a mechanism to probe the impact of architectural choices on the performance of various methods. Different architectural choices can be sensitive to the distribution of information across the feature components of a modality (e.g. how the information is distributed across pixels in an image). In this example we introduce structural equations that enable ablation studies that can independently isolate the role of (pointwise) mutual information from the distribution of information.

As discussed in Section 3.4, templates can control how the information is distributed among the components of each latent $\mathbf{z}_i$ while preserving the total mutual information between two modalities. This provides a mechanism for disentangling the effects of algorithmic (e.g. specific SSL objectives) from architectural choices (e.g. inductive biases in the model construction) by independently varying where the information is distributed in the data. The following structural equations incorporate modality-specific templates associated to a set of shared proto-latents representing common causes $\tilde{u}_k$ as well as a path for shared information from the proto-latents $\hat{\mathbf{u}}_j$ that are independent of the target latent $z_\theta$:

$$z_\theta = \sum_{k=1}^{N_\theta} \eta_k \tilde{u}_k \qquad \mathbf{z}_i = \sum_{k=1}^{N_\theta} \tilde{\rho}_{ik} \tilde{u}_k \mathbf{T}_{ik} + \sum_{j=1}^{N_u} \hat{\rho}_{ij} \hat{\mathbf{u}}_j, \tag{14}$$

The coefficients $\tilde{\rho}_{ik}$ and $\hat{\rho}_{ij}$ could either be independent hyperparameters or they could be parametrized as in Section 4.3, e.g. $\tilde{\rho}_{ik} = \beta^{-i} \tilde{\rho}$ and $\hat{\rho}_{ij} = \alpha^{-|i-j|} \hat{\rho}$.

## 5 Related Work

**Self-supervised learning and mutual information.** The relevance of mutual information to self-supervised learning, particularly contrastive learning, has been studied extensively. For example, the InfoNCE family of contrastive loss functions can be interpreted as bounds on the mutual information between representations (Oord et al., 2018; He et al., 2020; Caron et al., 2020). A range of work inspired by the InfoMax principle (Linsker, 1988) has argued that the MI between inputs and learned representations is an implicit target for multiview contrastive learning (Oord et al., 2018; Hjelm et al., 2019; Hénaff et al., 2020; Tsai et al., 2021). However, Tschannen et al. (2020) find that maximizing the MI alone is not sufficient for learning representations that are useful for downstream tasks,

stressing that the relation between estimated MI and representation quality depends strongly on both architecture choice and the form of the mutual information estimator used. More recent work (Tian et al., 2020; Wang et al., 2022; Wang & Isola, 2022; Rodríguez-Gálvez et al., 2023) explores this question in further depth. Our ability to generate realistic, complex datasets with known mutual information may enable further progress in determining the role of information maximization in self-supervised learning.

**Mutual information estimation and benchmarking.** Estimating mutual information from samples is challenging (McAllester & Stratos, 2020), especially in compelling real-world datasets and applications (Holmes & Nemenman, 2019; Gao et al., 2015; 2017). A rich body of literature covers a variety of mutual information estimation methods, ranging from traditional approaches based on histogram density or $k$-nearest neighbors (Pizer et al., 1987; Kozachenko & Leonenko, 1987; Kraskov et al., 2004) to neural estimators based on variational approaches (Belghazi et al., 2018; Song & Ermon, 2020b) or generative modeling (Ao & Li, 2022; Butakov et al., 2024). However, benchmarking the efficacy of these estimators on realistic datasets is entirely nontrivial. Most existing approaches are validated on multivariate normal distributions where mutual information is easily controllable, while recent work has explored simple transformations of these distributions to emulate properties of real data (Czyż et al., 2023a;b; Butakov et al., 2023). Our work replaces these simple transformations with a flexible bijective mapping learned through flow-based generative modeling (Chen et al., 2018b), enabling construction of highly realistic datasets with analytically tractable mutual information.

**Mutual information-preserving transforms.** Several generative models satisfy the mutual information preservation condition. Discrete-time normalizing flows with coupling layers (Rezende & Mohamed, 2015; Papamakarios et al., 2021) and Invertible Residual Networks (Behrmann et al., 2019) were among the first invertible (bijective) deep generative models. More recently, TARFLOW and STARFLOW (Zhai et al., 2024; Gu et al., 2025) have achieved very strong results on high resolution image generation. The ability of flow-based models to preserve mutual information has been employed in a variety of contexts, including mutual information estimation (Butakov et al., 2024) and developing alternative prescriptions for training flow models (Ardizzone et al., 2021). However, employing this property to develop realistic datasets with known mutual information remains a novel contribution of this work.

## 6 CONCLUSION

We present a new framework for generating realistic datasets with many modalities that are designed with known and controllable mutual information. Our dataset generation framework uses interpretable causal models with linear structural equations to construct correlated, normally-distributed latent variables with known mutual information. Blocks of components of these random variables are then fed into invertible (bijective) transformations that map the latent inputs into a realistic feature space while preserving the mutual information content. These realistic and nontrivial datasets enable numerous studies, including benchmarking studies of mutual information estimators. Critically, these datasets will be important for understanding and validating the role of mutual information in various multimodal self-supervised learning strategies, particularly as the number of modalities grows.

### REPRODUCIBILITY

To encourage reproducibility, we submit all code as supplementary material. Additionally, we describe the details of our example datasets, including hyperparameters chosen.

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

## A  PREDICTION ERROR FOR THE $\theta$ REGRESSION TASK

In Figure 7, we show the distribution of prediction errors $\theta - \hat{\theta}$ for two illustrative points along our MI distribution. The dataset with lower MI (in blue) exhibits a wider spread in errors, indicating decreased regression performance, while the dataset with higher MI (in orange) shows a sharper peak, indicating better regression performance. Moreover, both distributions are centered at 0, indicating that they are not biased in their estimations of $\theta$.

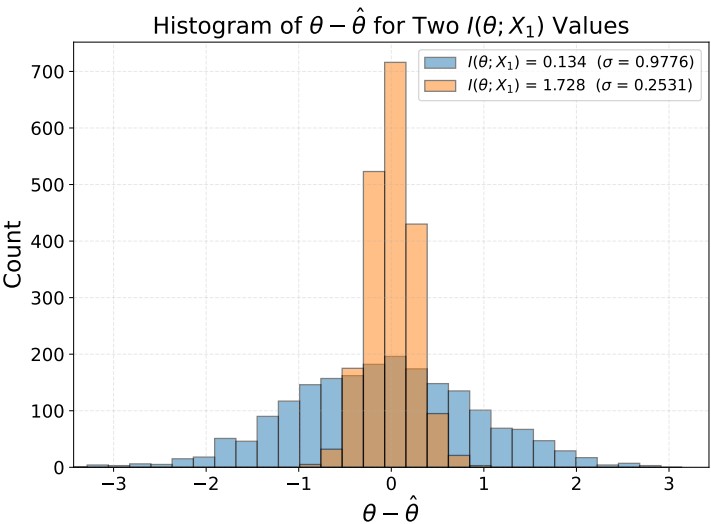

Figure 7: An example distribution of prediction error $\theta - \hat{\theta}$ for two representative MI values.

## B  ANALYTIC FORMULAE FOR COVARIANCE MATRICES AND MUTUAL INFORMATION

For a causal story where:

$$z_\theta = \sum_{k=1}^{N_\theta} \eta_k \cdot \tilde{u}_k$$

$$\mathbf{z}_i = \sum_{k=1}^{N_\theta} \tilde{\rho}_{ik} \cdot \tilde{u}_i + \sum_{j=1}^{N_u} \hat{\rho}_{ij} \cdot \hat{\mathbf{u}}_i \tag{15}$$

We can represent the covariance matrix for a bimodal case as follows:

$$\Sigma = \begin{bmatrix} a & r_1 \mathbf{1}^\top & r_2 \mathbf{1}^\top \\ r_1 \mathbf{1} & (b-o)\mathbf{I}_d + o\mathbf{J}_d & (f-p)\mathbf{I}_d + p\mathbf{J}_d \\ r_2 \mathbf{1} & (f-p)\mathbf{I}_d + p\mathbf{J}_d & (c-q)\mathbf{I}_d + q\mathbf{J}_d \end{bmatrix} \tag{16}$$

where $\mathbf{I}_d$ and $\mathbf{J}_d$ are the $d \times d$ identity and matrix of ones, respectively, and:

- $a = \sum_{k=1}^{N_\theta} \eta_k^2$

- $r_1 = \sum_{k=1}^{N_\theta} \eta_k \tilde{\rho}_{1k} \qquad r_2 = \sum_{k=1}^{N_\theta} \eta_k \tilde{\rho}_{2k}$

- $o = \sum_{k=1}^{N_\theta} (\tilde{\rho}_{1k})^2 \qquad q = \sum_{k=1}^{N_\theta} (\tilde{\rho}_{2k})^2$

- $p = \sum_{k=1}^{N_\theta} \tilde{\rho}_{1k} \tilde{\rho}_{2k}$

- $b = o + \sum_{j=1}^{N_u} (\hat{\rho}_{1j})^2 \qquad c = q + \sum_{j=1}^{N_u} (\hat{\rho}_{2j})^2$

- $f = p + \sum_{j=1}^{N_u} \hat{\rho}_{1j} \hat{\rho}_{2j}$

Because $\mathbf{I}_d$ and $\mathbf{J}_d$ commute, they can be simultaneously diagonalized. Thus, for a block like

$$\Sigma_{11} = (b - o)\mathbf{I}_d + o\mathbf{J}_d, \tag{17}$$

the eigenvalues are:

- $b + (d-1)o$ (multiplicity 1, for $\mathbf{1}_d$)
- $b - o$ (multiplicity $d-1$, for vectors orthogonal to $\mathbf{1}_d$).

Therefore,

$$|\Sigma_{11}| = (b-o)^{d-1}[b + (d-1)o] \tag{18}$$

$$|\Sigma_{22}| = (c-q)^{d-1}[c + (d-1)q . \tag{19}$$

By the matrix determinant lemma and Schur complement, the determinant of a block matrix $\Gamma_{ij}$ can be written as

$$|\Gamma_{ij}| = |\Sigma_{jj}| \cdot \left| \Sigma_{ii} - \Sigma_{ij} \Sigma_{jj}^{-1} \Sigma_{ji} \right| , \tag{20}$$

and thus

$$|\Gamma_{\theta 1}| = (b-o)^{d-1} \left[ a(b + (d-1)o) - dr_1^2 \right] \tag{21}$$

$$|\Gamma_{12}| = \left[ (b-o)(c-q) - (f-p)^2 \right]^{d-1} \left[ (b + (d-1)o)(c + (d-1)q) - (f + (d-1)p)^2 \right] . \tag{22}$$

Some examples of closed-form equations using these terms are

$$I(\theta; Z_1) = -\frac{1}{2} \log \left( 1 - \frac{dr_1^2}{a[b + (d-1)o]} \right) \tag{23}$$

$$I(\theta; Z_2) = -\frac{1}{2} \log \left( 1 - \frac{dr_2^2}{a[c + (d-1)q]} \right) \tag{24}$$

$$I(Z_1; Z_2) = \frac{d-1}{2} \log \left( \frac{(b-o)(c-q)}{(b-o)(c-q) - (f-p)^2} \right)$$
$$+ \frac{1}{2} \log \left( \frac{[b + (d-1)o][c + (d-1)q]}{(b + (d-1)o)(c + (d-1)q) - (f + (d-1)p)^2} \right) . \tag{25}$$

These equations have been verified against the numerical calculations.

