# OpenReview forum: "Multimodal Datasets with Controllable Mutual Information"
_ICLR.cc/2026/Conference — Submitted to ICLR 2026_

### Official Review · Reviewer_RwuG · 2025-10-24

**Soundness:** 2
**Presentation:** 2
**Contribution:** 2
**Rating:** 2
**Confidence:** 4

**Summary:**

This paper introduces a framework for generating multimodal datasets where the mutual information between modalities is measurable and controllable. This is useful for lots of works which study the mutual information between modalities and labels in multimodal training dynamics.

**Strengths:**

- The framework for generating controllable mutual information seems correct and insightful.
- There are a lot of important use cases for this: so many multimodal works look at training based on mutual information. Having it controlled synthetically would be a powerful and useful testbed for that research, and could lead to an important breakthrough in that field.

**Weaknesses:**

Unfortunately, I don't think this paper did quite enough to justify that this framework could be used for the strengths I outlined above. A few key points:
- What is the practical utility of this work? You could for example show that your dataset provides training transfer to realistic environments with mutual information-dependent training methods. But without that, how do we know the value of the data you generate with your method?
- If there isn't transfer of performance or key insights from training, what insights can you get by studying models on this dataset, and will those insights transfer to models' behavior on real world datasets? If so, this could be a useful prototyping tool that allows people to run and understand experiments theoretically before doing computationally expensive and confusing training runs on messy real world data. For example, can you show that some findings from prior work are mirrored in your setting, and can be ascertained quickly and reliably, whereas training on a full real world dataset would be costly and noisy?

**Questions:**

- How would you simulate handle modality imbalances? Where some data have missing modalities or you have large amounts of unimodal data.
- I didn't understand the black hole example. Could you clarify the motivation?

---

> ### Author Response · Authors · 2025-11-26
> **Rebuttal Part 1/2**
>
> We thank the reviewer for their time and helpful comments. RwuG notes that our framework is **“correct and insightful”**, has **“important use cases”**, and **“could lead to an important breakthrough in that field”**. We address specific comments below.
>
> > What is the practical utility of this work? … how do we know the value of the data you generate with your method? … what insights can you get by studying models on this dataset, and will those insights transfer to models' behavior on real world datasets?
>
> The primary purpose of this work is to construct datasets that enable us to empirically evaluate the role of mutual information in self-supervised learning with realistic datasets, which is an ongoing challenge in the field. Since it is notoriously difficult to estimate MI from high-dimensional real-world datasets, we have two primary use cases in mind:
>
> **Nontrivial benchmarking for mutual information estimators:** A variety of MI estimators for real-world data have been developed in existing literature but are typically only validated on synthetic data or simple distributions for which the MI is easily calculable. Our dataset generating framework provides the tools to generate significantly more realistic datasets with known MI than these existing benchmarking techniques while preserving MI.
>
> **Studies of various SSL methods:** Studying SSL with realistic datasets with known MI has the potential to greatly improve our collective understanding of these methods, e.g., what is the role of shared/similar features vs. shared mutual information not included in the input features, and how does this behavior scale as new modalities are added?
>
> During the rebuttal period, we have added several experiments that demonstrate how our framework can be used in practice. In particular, as an example of a benchmark study, **we evaluate eight leading MI estimators on 10 datasets generated using our framework and find that all estimators we test fall within about 30% of the true MI**, though their accuracies vary (Figure 3, Section 4.1 of the new revision). Additionally, we have added results for a downstream regression task that shows that the **RMSE of a fully supervised estimator for a target value decreases monotonically as the MI between the input data and target value increases** (Figure 4, Section 4.2). We hope that these additional empirical studies clarify the use and value of our framework.
>
>
> > …this could be a useful prototyping tool that allows people to run and understand experiments theoretically before doing computationally expensive and confusing training runs on messy real world data. For example, can you show that some findings from prior work are mirrored in your setting, and can be ascertained quickly and reliably, whereas training on a full real world dataset would be costly and noisy?
>
> This is a very interesting suggestion. We can envision a scenario where an MI estimator is applied on a real world dataset, then we can use our framework to emulate this dataset’s MI properties and train on this emulated dataset to predict what will happen on the noisier real-world dataset. This would allow one to explore hyper-paramter tuning or do ablation studies related to training with simulated data. This can be particularly helpful in a low real-data regime.

---

> > ### Author Response · Authors · 2025-11-26
> > **Rebuttal Part 2/2**
> >
> > > How would you simulate handle modality imbalances? Where some data have missing modalities or you have large amounts of unimodal data.
> >
> > Our framework simultaneously generates data for multiple modalities from a joint distribution with explicitly calculable mutual information, after which modality-specific sub-sampling can be performed to simulate modality imbalances. Imbalanced sampling will certainly impact the training and performance of SSL models, and can be studied using our framework by randomly dropping samples from a given modality. While this was not part of our initial scope, it provides another example of how our framework can be used to perform ablation studies that impact SSL in practice.
> >
> > > I didn't understand the black hole example. Could you clarify the motivation?
> >
> > The black hole example is intended as a minimal, scientifically-motivated example for the target parameters $\theta$ (in this case, the scalar-valued mass of the black hole). It was also meant to provide a pedagogical example for our structured causal model that treats multiple causal factors asymmetrically. We envision a scenario in which the input data come in two modalities: image data from a ground-based telescope and images from a space-based telescope, represented as $x_1,x_2$. In our imagined scenario, the black hole itself (denoted $\tilde{u_1}$ in the text) influences the data from both modalities $x_1$ and $x_2$ (since it’s the subject of both images). In contrast, the atmospheric effects (denoted $\tilde{u_2}$) only influence the data from ground-based telescopes $x_2$. We then show in Table 2 that there is no mutual information between the atmospheric effects $\tilde{u_2}$ and the space-based telescope data as expected.
> >
> > [1] Lee & Rhee, 2024. A Benchmark Suite for Evaluating Neural Mutual Information Estimators on Unstructured Datasets.

---

### Official Review · Reviewer_Yb4Z · 2025-10-27

**Soundness:** 2
**Presentation:** 2
**Contribution:** 2
**Rating:** 2
**Confidence:** 3

**Summary:**

The paper proposes a framework for generating synthetic data with controlled mutual information using flow-based generative models and a structured casual framework. The illustration of the data generation pipeline is followed by two brief discussions of the example usage in generating synthetic data for different underlying causal structures and scales of modalities.

**Strengths:**

- The paper is well-motivated, as there has been an emerging interests in multimodal learning from an information-theoretic approach, and this paper provides a well-suited, controlled testbed for such types of research;
- The proposed data generation pipeline is novel, well-documented and clearly explained;

**Weaknesses:**

- One major limitation of this work is the lack of empirical evaluation, neither qualitative evaluation (e.g. Figure 2, which the paper also acknowledges that "there is no clear visual connection between these pairs of images") nor quantitative evaluation. This makes it **very hard to verify the correctness** of the proposed framework. In particular, the reviewer does not agree with the claim that "our framework allows us to state unequivocally that these high-dimensional, complex datasets have a specific amount of mutual information" due to this lack of empirical evidence. The paper also does not give any empirical evaluation using the synthetic data generated from the proposed pipeline, so **the claims about the practical utility is also not testified**.

**Questions:**

- The reviewer strongly recommends adding more empirical evaluation of the proposed pipeline to show (1) the correctness (e.g. either qualitatively or quantitatively verify the generated data are indeed correlated by the given mutual information) and (2) the utility via a minimal set of evaluations of existing information-theoretic multimodal learning approaches on the generated data, followed by analysis on the results and potential insights that can be meaningful towards multimodal learning research from an information-theoretic perspective

---

> ### Author Response · Authors · 2025-11-26
> **Rebuttal**
>
> We thank the reviewer for their time and thoughtful comments. Yb4Z notes that our paper is **“well-motivated”** and our approach is **“novel, well-documented, and clearly explained”**. We address specific comments below.
>
> > One major limitation of this work is the lack of empirical evaluation … This makes it very hard to verify the correctness of the proposed framework. … The reviewer strongly recommends adding more empirical evaluation of the proposed pipeline to show (1) the correctness (e.g. either qualitatively or quantitatively verify the generated data are indeed correlated by the given mutual information)
>
> We agree with the reviewer that incorporating more empirical results into the paper would be valuable to illustrate the capabilities of our framework. However, because our dataset generation method relies on the well-established theorem that MI is invariant to bijective transformations, we stress that our method must preserve MI by construction and the correctness of our framework stands on theoretical grounds. Furthermore, we emphasize that empirical MI estimation is notoriously difficult and empirical MI estimators vary substantially in accuracy when evaluated on datasets with ground-truth MI (e.g., [1]), leading us to view their role as supporting rather than validating the correctness of our framework.
>
> Nevertheless, we have added several experiments that demonstrate how our framework can be used in practice. In particular, as an example of a benchmark study, **we evaluate eight leading MI estimators on 10 datasets generated using our framework and find that all estimators we test fall within about 30% of the true MI**, though their accuracies vary (Figure 3, Section 4.1 of the new revision). Additionally, **we have added results for a downstream regression task that shows that the RMSE of a fully supervised estimator for a target value decreases monotonically as the MI between the input data and target value increases** (Figure 4, Section 4.2). We hope that these additional empirical studies clarify the use and value of our framework.
>
> > The paper also does not give any empirical evaluation using the synthetic data generated from the proposed pipeline … The reviewer strongly recommends adding more empirical evaluation of the proposed pipeline to show .. (2) the utility via a minimal set of evaluations of existing information-theoretic multimodal learning approaches on the generated data
>
> One of the mutual information estimators in our new experiment (Section 4.1) uses the InfoNCE objective (Figure 3, orange points), which is commonly used in self-supervised learning. Our results reinforce that the InfoNCE objective is connected to mutual information, as expected, and lay the groundwork for additional studies of SSL through the lens of mutual information with our framework.
>
>
> [1] Lee & Rhee, 2024. A Benchmark Suite for Evaluating Neural Mutual Information Estimators on Unstructured Datasets.

---

### Official Review · Reviewer_LU2w · 2025-10-27

**Soundness:** 3
**Presentation:** 3
**Contribution:** 2
**Rating:** 4
**Confidence:** 3

**Summary:**

The paper proposes a novel framework for generating high-dimensional multimodal datasets with controllable mutual information. By using flow-based generative models, the method ensures that mutual information between latent variables is preserved, providing a theoretical foundation. The paper also designs a structured causal framework to generate correlated latent variables, derive closed-form analytical formulas for mutual information, and provide examples of synthetic multimodal datasets illustrating different causal and correlation patterns.

**Strengths:**

1. The paper proposes a framework for generating high-dimensional multimodal data with controllable mutual information, which is rarely achieved in existing public datasets or prior methods.

2. By leveraging flow-based generative models, the approach ensures that the generated data preserves mutual information between latent variables, providing a theoretical foundation.

**Weaknesses:**

1. All experiments are conducted solely on CIFAR-10 image data, without demonstrating results on real multimodal datasets (e.g., CMU-MOSI, CMU-MOSEI, or video-text-audio combinations).

2. The paper does not evaluate the generated data on downstream tasks (e.g., regression or classification), making it difficult to quantitatively assess its contribution. It also lacks direct comparison with existing mutual information estimators or multimodal SSL approaches.

3. Some concepts (e.g., the template and flow matching) are not intuitive to non-expert readers, and overall readability could be improved. Moreover, the paper is limited to 8 pages, whereas the ICLR 2026 initial submission allows up to 9 pages.

**Questions:**

1. How does the generated data impact performance on downstream tasks, such as regression or classification?

2. Could the authors provide a comparison of their approach with existing mutual information estimators or multimodal SSL methods to better contextualize the contributions?

---

> ### Author Response · Authors · 2025-11-26
> **Rebuttal**
>
> Thank you for your thoughtful review of our paper, and for noting that our work pairs a **“theoretical foundation”** with a result that **“is rarely achieved in existing public datasets or prior methods”**. Below, we have expanded on your comments and questions:
>
> > W1: All experiments are conducted solely on CIFAR-10 image data, without demonstrating results on real multimodal datasets (e.g., CMU-MOSI, CMU-MOSEI, or video-text-audio combinations).
>
> In this work, we define a modality as a distinct source of data used as an input for an ML model. Modalities can vary between the structurally heterogeneous (e.g. image vs. text vs. 3D point cloud) and the structurally homogeneous (e.g. IR heatmap vs. RGB image vs. segmentation maps, all of which have the structure of a dense 2D image). This definition has been broadly adopted in large-scale multimodal models such as [1] that treat structurally similar inputs such as RGB, depth maps, surface normals, and semantic segmentation as distinct modalities. We also note that flow matching has been implemented on diverse modalities such as audio [2] and video [3], so it would be straightforward to extend our framework to these modalities. While we did not show explicit examples of these other modalities in this work, doing so would not significantly alter the presentation of the method or its central conclusions, nor would it offer new fundamental insights.
>
> > W2/Q1: The paper does not evaluate the generated data on downstream tasks (e.g., regression or classification), making it difficult to quantitatively assess its contribution. … How does the generated data impact performance on downstream tasks, such as regression or classification?
>
> We now include new results (see Fig. 4) showing that RMSE of a fully supervised estimator for a scalar prediction target (denoted $\theta$ in the paper) decreases monotonically as the ground-truth MI between the input data and target value increases.
>
> > W2/Q2: [The paper] also lacks direct comparison with existing mutual information estimators or multimodal SSL approaches … Could the authors provide a comparison of their approach with existing mutual information estimators or multimodal SSL methods to better contextualize the contributions?
>
> We agree with the reviewer that incorporating more empirical results into the paper would be valuable to illustrate the capabilities of our framework. Because our dataset generation method relies on the well-established theorem that MI is invariant to bijective transformations, we stress that our method must preserve MI by construction. Nevertheless, we have added several experiments that demonstrate how our framework can be used in practice. In particular, as an example of a benchmark study, **we evaluate eight leading MI estimators on 10 datasets generated using our framework and find that all estimators we test fall within about 30% of the true MI**, though their accuracies vary (Figure 3, Section 4.1 of the new revision). Additionally, we have added results for a downstream regression task that shows that the **RMSE of a fully supervised estimator for prediction target $\theta$ decreases monotonically as the MI between the input data and target value increases** (Figure 4, Section 4.2).
>
> One of the mutual information estimators in our new experiment (Section 4.1) uses the InfoNCE objective, which is commonly used in self-supervised learning. Our results (Table 1 and Figure 3, orange points) show that **MI estimates from InfoNCE are well-correlated (r=0.991) and close to (RMSE=0.2) our datasets’ ground-truth MI**. This reinforces that the InfoNCE objective is connected to mutual information, as expected, and lays the groundwork for additional studies of SSL through the lens of mutual information with our framework.
> We hope that these additional empirical studies clarify the use and value of our framework.
>
> > W3: Some concepts (e.g., the template and flow matching) are not intuitive to non-expert readers, and overall readability could be improved.
>
> In the new revision, we have added Section 3.4 to describe templates in much greater detail. We thank the reviewer for this suggestion.
>
>
> [1] Mizrahi et al., 2023. 4M: Massively Multimodal Masked Modeling.
>
> [2] Liu et al., 2024. Generative Pre-training for Speech with Flow Matching.
>
> [3] Jin, et al., 2025. Pyramidal Flow Matching for Efficient Video Generative Modeling.

---

### Official Review · Reviewer_grni · 2025-10-30

**Soundness:** 2
**Presentation:** 3
**Contribution:** 3
**Rating:** 2
**Confidence:** 3

**Summary:**

This paper proposes a framework for generating synthetic multimodal datasets with explicitly controllable MI between modalities. The method combines causal latent-variable construction with flow-based generative models that preserve MI under bijective mappings. The authors claim this provides a testbed for studying multimodal SSL and benchmarking mutual information estimators.

**Strengths:**

Although data generation is not my primary area of expertise, this work appears to address a genuinely underexplored and important problem: constructing realistic high-dimensional multimodal datasets with analytically tractable and controllable mutual information, which could enable systematic evaluation of self-supervised learning methods and mutual information estimators. The theoretical development is simple and clear. The use of flow-based generative models to maintain information structure across high-dimensional modalities is conceptually elegant and technically well-motivated.

**Weaknesses:**

The main limitation of this paper lies in the absence of empirical validation. While the framework is theoretically elegant, the paper does not demonstrate that the generated datasets are practically useful for their intended purposes, such as evaluating self-supervised learning methods or mutual information estimators. The examples provided are purely illustrative and rely on analytic expressions rather than experiments that confirm controllability or MI preservation in practice. Moreover, the claim of producing “realistic multimodal data” is overstated: using CIFAR-10 class-conditioned flows as a proxy for distinct modalities is a weak approximation of genuine multimodality (e.g., image–text, video–audio, etc.), and it remains unclear whether the generated samples exhibit meaningful cross-modal relationships. The reliance on linear-Gaussian causal structures, while analytically convenient, limits the generality of the approach for more complex, nonlinear dependencies in real-world multimodal settings. The paper would also benefit from quantitative experiments comparing analytical MI values with empirical estimates obtained via neural MI estimators to substantiate its proposed utility.

**Questions:**

1. Can you provide empirical evidence that the generated datasets preserve the specified mutual information after flow transformations?
2. Have you tested any self-supervised learning methods to demonstrate that controllable MI affects downstream performance as intended?
3. Does your linear-Gaussian setup generalize to nonlinear or non-Gaussian latent dependencies?
4. How scalable is the framework to higher-dimensional data (e.g. video or time series) or more modalities?
5. Have you evaluated how well existing mutual information estimators recover the known MI values on your generated datasets?

---

> ### Author Response · Authors · 2025-11-26
> **Rebuttal**
>
> We thank the reviewer for their time and insightful comments. grni notes that **“this work appears to address a genuinely underexplored and important problem”**, **“the theoretical development is simple and clear”**, and **“the use of flow-based generative models [...] is conceptually elegant and technically well-motivated.”**
> We address specific comments below.
>
> > W1: “The main limitation of this paper lies in the absence of empirical validation.”
>
> > Q1: Can you provide empirical evidence that the generated datasets preserve the specified mutual information after flow transformations?
>
> > Q5: Have you evaluated how well existing mutual information estimators recover the known MI values on your generated datasets?
>
> We agree with the reviewer that incorporating more empirical results into the paper would be valuable to illustrate the capabilities of our framework. Because our dataset generation method relies on the well-established theorem that MI is invariant to bijective transformations, we stress that our method must preserve MI by construction. Nevertheless, we have added several experiments that demonstrate how our framework can be used in practice. In particular, as an example of a benchmark study, **we evaluate eight leading MI estimators on 10 datasets generated using our framework and find that all estimators we test fall within about 30% of the true MI**, though their accuracies vary (Figure 3, Section 4.1 of the new revision). Additionally, we have added results for a downstream regression task showing that **RMSE of a fully supervised estimator for a target parameter (denoted $\theta$ in the paper) decreases monotonically as the MI between the input data and target value increases** (Figure 4, Section 4.2). We hope that these additional empirical studies clarify the use and value of our framework.
>
>
> > Q2: Have you tested any self-supervised learning methods to demonstrate that controllable MI affects downstream performance as intended?
>
> One of the mutual information estimators in our new experiment (Section 4.1) uses the InfoNCE objective, which is commonly used in self-supervised learning. Our results (Table 1 and Figure 3, orange points) show that **MI estimates from InfoNCE are well-correlated (r=0.991) and close to (RMSE=0.2) our datasets’ ground-truth MI**. This reinforces that the InfoNCE objective is connected to mutual information, as expected, and lays the groundwork for additional studies of SSL through the lens of mutual information with our framework.
>
> > Q3: The reliance on linear-Gaussian causal structures, while analytically convenient, limits the generality of the approach for more complex, nonlinear dependencies in real-world multimodal settings … Does your linear-Gaussian setup generalize to nonlinear or non-Gaussian latent dependencies?
>
> The choice of linear-Gaussian causal structure is not intrinsic to our framework. We employ it only in order to render ground-truth mutual information easily-computable. The linear relationships and the Gaussianity of generated latents are not preserved through the flows (as flows are highly nonlinear); however, the mutual information is preserved. If the generating causal process is not linear-Gaussian, our framework will still preserve mutual information by construction.
>
>
> > Q4: How scalable is the framework to higher-dimensional data (e.g. video or time series) or more modalities?
>
> While we focused on image-based modalities in this work, flow matching can be applied to preserving MI for any fixed-length continuous-valued modality (e.g. video [1] or audio [2]). We emphasize that our theoretical framework is modality- and distribution-agnostic and designed for users to plug in their own trained flows (bijective generative models) trained on data relevant to their problem setting. Further, Section 4.3 (Example 3) illustrates how our model scales easily to 10+ modalities – indeed, a central motivator of our work is to help understand how existing SSL methods scale with increasing numbers of modalities.
>
> [1] Jin, et al., 2025. Pyramidal Flow Matching for Efficient Video Generative Modeling.
>
> [2] Liu et al., 2024. Generative Pre-training for Speech with Flow Matching.

---

### Author Response · Authors · 2025-11-26
**Overall Rebuttal Comment**

We are grateful to the reviewers for their thoughtful and constructive feedback. We are glad that the reviewers have found our paper to be **conceptually elegant, novel, and insightful** [grni, Yb4Z, RwuG] with results that are **“rarely achieved in existing public datasets or prior methods”** [LU2w]. We have updated our submission to incorporate their feedback, which we believe has strengthened our submission.

We agree with several of the reviewers' comments that incorporating more empirical results into the paper would be valuable to illustrate the capabilities of our framework. We have added several experiments that demonstrate how our framework can be used in practice. In particular, as an example of a benchmark study, we evaluate eight leading MI estimators on 10 datasets generated using our framework . Additionally, we have added results for a downstream regression task. These can be found in subsection 4.1. We hope that these new results address those concerns.

In addition, we added a new subsection 3.4 expanding on the description of Templates and added a new subsection 4.4 describing another example use case of our framework to conduct ablation studies for multimodal SSL. There are also several other small clarifications throughout the paper. Thank you.

We invite the reviewers to examine the updated submission and welcome any additional questions, comments, or suggestions for further improvement. If you wish to see the tracked changes, you can view the anonymized pdf linked here: https://anonymous.4open.science/r/ICLR-2026-Submission-E870/Diff_of_ICLR_2026__Multimodal_Datasets_with_Controllable_MI.pdf. We also respond to each individual reviewer in separate responses.

---

### Meta-Review · Area_Chair_m62N · 2025-12-31

**Summary:**

The reviewers’ main concerns are the lack of sufficient empirical validation, the limited generality of the experimental setting, and the unclear practical utility of the framework. The current experiments are restricted to image-only and largely synthetic settings, which do not convincingly support claims about cross-modality generalization or real-world relevance. The rebuttal provides mainly conceptual or theoretical arguments, but does not address the need for additional empirical evidence, especially in high-dimensional or heterogeneous scenarios. As a result, the key concerns raised by the reviewers remain unresolved.

**Reviewer Concerns:**

Several reviewers (grni and LU2w) raise concerns regarding the insufficiency of the experimental evaluation. In particular, the current version only involves image-based data, which may not be sufficient to support the generality of the claims. They suggest providing empirical results across heterogeneous modalities, such as image–text settings.

In the rebuttal, the authors respond by appealing to their definition of “modalities,” emphasizing structurally homogeneous cases (e.g., IR heatmaps vs. RGB images). While this clarification is conceptually reasonable, it does not fully address the reviewers’ concern. In the current experiments, both “modalities” are still images and differ mainly by being conditioned on different class labels. This does not convincingly demonstrate cross-modality generality in the sense intended by the reviewers, and therefore does not sufficiently support the broader claims.

----

Reviewer Yb4Z further points out a major limitation regarding the lack of empirical evaluation, especially for high-dimensional data. The rebuttal argues that mutual information is invariant under bijective transformations, and thus that the results extend theoretically. While this argument is correct at a theoretical level, there remains a substantial gap to real-world applications, where one may learn (or approximate) mappings from low-dimensional latent variables to high-dimensional observations, in this respect I agree that additional empirical evidence is needed to support the claims.

----

Reviewer RwuG mainly questions the practical utility of the proposed framework. The rebuttal offers two main responses. First, it is suggested that the framework enables nontrivial benchmarking of mutual information estimators by generating more “realistic” datasets. However, given that both the latent variable distributions and the underlying DAG structure are still fully synthetic and known by construction, it is unclear how much additional realism this setting provides, or how it addresses the reviewer’s concern about practical relevance.

Second, the authors mention potential applications to studying SSL. However, the work does not provide convincing results on SSL representations to substantiate this claim. Moreover, there is already a body of work in the causal representation learning community that studies the relationship between shared latent variables and multimodal representation learning, both theoretically and empirically. Without concrete experiments or clearer differentiation from this prior work, this point remains insufficiently supported.

Overall, while the rebuttal clarifies some conceptual aspects of the framework, it does not fully resolve the reviewers’ concerns about experimental sufficiency, generality beyond image-only settings, and practical utility. Additional empirical validation, especially in genuinely heterogeneous or high-dimensional settings and with clearer demonstrations of downstream relevance, would be necessary to substantiate the claims.

**Reviewer Scores:**

grni:  Core concern about limited experimental sufficiency and lack of cross-modality validation not addressed.

LU2w: Rebuttal does not resolve the concern about generality beyond image-only settings.

Yb4Z:  Theoretical justification does not address the lack of empirical results in high-dimensional settings.

RwuG: Practical utility remains insufficiently demonstrated.

Overall: The rebuttal does not materially resolve the main concerns raised by reviewers, no positive score changes are expected.

---

### Decision · Program_Chairs · 2026-01-26

Reject